# A Physically Interpretable Rice Field Extraction Model for PolSAR Imagery

**Ji Ge** [1,2,3], **Hong Zhang** [1,2,3,*], **Lu Xu** [1,2], **Chunling Sun** [1,2,3], **Haoxuan Duan** [1,2,3], **Zihuan Guo** [1,2,3] **and Chao Wang** [1,2,3]

1   Key Laboratory of Digital Earth Science, Aerospace Information Research Institute, Chinese Academy of Sciences, Beijing 100094, China
2   International Research Center of Big Data for Sustainable Development Goals, Beijing 100094, China
3   College of Resources and Environment, University of Chinese Academy of Sciences, Beijing 100049, China
*   Correspondence: zhanghong@radi.ac.cn

**Abstract:** Reliable and timely rice distribution information is of great value for real-time, quantitative, and localized control of rice production information. Synthetic aperture radar (SAR) has all-weather and all-day observation capability to monitor rice distribution in tropical and subtropical areas. To improve the physical interpretability and spatial interpretability of the deep learning model for SAR rice field extraction, a new SHapley Additive exPlanation (SHAP) value-guided explanation model (SGEM) for polarimetric SAR (PolSAR) data was proposed. First, a rice sample set was produced based on field survey and optical data, and the physical characteristics were extracted using decomposition of polarimetric scattering. Then a SHAP-based Physical Feature Interpretable Module (SPFIM) combing the long short-term memory (LSTM) model and SHAP values was designed to analyze the importance of physical characteristics, a credible physical interpretation associated with rice phenology was provided, and the weight of physical interpretation was combined with the weight of original PolSAR data. Moreover, a SHAP-guided spatial interpretation network (SSEN) was constructed to internalize the spatial interpretation values into the network layer to optimize the spatial refinement of the extraction results. Shanwei City, Guangdong Province, China, was chosen as the study area. The experimental results showed that the physical explanation provided by the proposed method had a high correlation with the rice phenology, and spatial self-interpretation for finer extraction results. The overall accuracy of the rice mapping results was 95.73%, and the kappa coefficient reached 0.9143. The proposed method has a high interpretability and practical value compared with other methods.

**Keywords:** rice field; interpretability; PolSAR; deep learning

## 1. Introduction

Rice is one of the most important food crops in the world and serves as a major food source for over half of the world's population [1]. In 2018, the statistics from the Food and Agriculture Organization (FAO) of the United Nations show that rice accounts for more than 12% of global cultivated land [2]. Considering the protection of the ecological environment and the development of rice crops, it is essential to monitor rice production and accurately detect the distribution of rice fields. To alleviate the problem of optical sensor images being affected by clouds, synthetic aperture radar (SAR) with the characteristics of large-scale, all-weather, and all-day observation has proven to be an effective tool for mapping rice extent.

Considering that the full polarimetric SAR (PolSAR) is sensitive to crop geometric structure and water content changes, many scholars have used the physical parameters provided by PolSAR images to conduct crop classification research in the past decades [3,4]. Compared with the single-polarization SAR that can only provide backscatter coefficients,

PolSAR provides the phase information as well as the intensity information of four polarizations, which makes it possible to characterize the backscattering mechanism [5]. The effectiveness of the polarization ratio of the backscattering coefficient [6], Pauli decomposition [7], Freeman–Durden decomposition [8–10], the Cloude–Pottier decomposition [10,11], the Yamaguchi decomposition [11], the van Zyl decomposition and the Krogager decomposition [12] have been demonstrated by existing research. With the thriving of PolSAR data, the crop classification and rice recognition studies were conducted using different data acquisition schemes [13–15]. Comparative studies of PolSAR and other SAR schemes were also carried out, which demonstrated the superiority of full polarimetric SAR over dual and compact polarimetric SAR data [16–18]. The expansion of SAR data reserves in recent years has provided good opportunities for rice monitoring, which utilized the evolution information of scattering mechanisms during the whole growth cycle [19–21]. Meanwhile, thanks to the fast development of big data processing methods, the rice classification methods were also evolved from the threshold-based classifier [22] to the machine learning method [23–25], and now to the deep learning models [26,27].

The above studies mainly used the physical feature parameters extracted from PolSAR data and traditional classifiers (e.g., threshold method, SVM, and Random Forest) to draw crop maps, confirming the great potential of physical features of PolSAR in crop classification. However, due to the limitations of classification algorithms, it is difficult to achieve a balance in terms of high accuracy, automation, and big data processing.

In recent years, given that the state-of-the-art SAR satellites with high temporal resolution provide continuous observations of rice paddies, and that the ability of deep learning to simulate both nonlinear and autoregressive phenomenon is effective for learning the phenological changes in crops, various deep learning methods have been applied to paddy rice extraction [28–43]. Existing studies have demonstrated the applicability of deep learning models to SAR data and their comparative advantages over classical machine learning methods. However, these studies have not been designed to adequately consider the variability of rice scattering mechanisms across different phenological periods. In contrast, studies using classical machine learning methods tend to introduce a priori knowledge, such as phenological information, to improve their extraction performance. The physical scattering mechanism information contained in PolSAR data is highly correlated with the crop growth process and is helpful to clarify the decision progress of classification algorithms, and only a few images are needed to accurately extract crop information [6,10,11].

Deep learning methods can extract valid abstract features from data, but users cannot intuitively understand these features, not to mention the analysis of their contribution to the results. In addition, erroneous results from neural networks can only be optimized by changing the network structure or aimlessly adjusting the dataset and cannot be improved in a targeted manner. As a result, there is a growing interest in the interpretability of deep learning models [44].

Researchers have identified transparency, comprehensibility, and interpretability as necessary conditions for interpretable deep learning models. Established deep learning interpretation methods can be mainly divided into post-hoc explanation methods [45–51] and self-explanatory methods [52–58]. Post-hoc explanations are stand-alone methods aimed at explaining an already trained and fixed target model. Interpretation based on feature importance is an important post-hoc explanatory method for explaining model predictions. Ribeiro et al. proposed Local Interpretable Model-Agnostic Explanations (LIME) [47], which uses interpretable models (e.g., linear models and decision trees) to locally approximate the predictions of a target black box model, but the interpretation results are not stable enough, and the results of repeated interpretations under the same conditions may be completely different. Lundberg and Lee proposed an additional interpretation method called SHAP which was based on the Shapley values [48]. The Shapley value is a solution concept in cooperative game theory. Unlike LIME, the SHAP interpretation satisfies three intuitive theoretical properties, namely local accuracy, missingness, and consistency. However, computing the exact Shapley value has exponential time complexity.

The use of sampling and weighted regression (Kernel SHAP), a modified backpropagation step (DeepSHAP) [48], the use of summed expectations [49] or making assumptions about the underlying data structure [59] are required to approximate Shapley values. All these solutions provide explanations of the features importance based on existing models, but fail to provide a physical-level understanding or have any impact on the model. On the other hand, self-explanatory methods integrate the interpretation generation module into their own architecture and have an impact on the training and prediction of the model, but may result in lower model performance than traditional deep learning models. It is important to note that both of the two types of explanatory methods are designed for the deep learning models that driven by natural optical data and may fail to incorporate the characteristics of SAR images.

At present, research on the interpretability of SAR deep learning algorithms is in its infancy [60–63]. In 2021, Shendryk et al. explained the contribution of different inputs to machine learning models for predicting sugarcane yield using SHAP [60]. In 2022, Al-Najjar et al. employed SHAP to measure the impact, interaction, and correlation of landslide condition factors in RF and SVM [61]. In addition, SHAP was applied to visualize and interpret the results of deep learning SAR oil slick detection [63]. In these studies, algorithms such as SHAP were used only for post-hoc interpretation analysis of model results and failed to fully incorporate the characteristics of SAR images. Since SAR images are different from traditional natural images in that they are characterized by large spatial scales and complex physical features, making the process more complex. Considering the limitations of deep learning interpretation methods and the trend of coupling SAR image characteristics with deep learning algorithms, we focus on "understanding", i.e., being able to understand why the model makes such decisions, from the perspective of SAR image characteristics. With regard to understanding, visual analysis of physical features and interpretation of spatial dimensions is preferable for explaining neural networks in the context of SAR applications and promising a balance between algorithmic transparency and intelligence. To this end, we propose a SHAP value-guided interpretable rice field extraction method (SGEM) using PolSAR data. The main contributions of this paper are as follows:

(1) The SHAP-based Physical Feature Interpretable Module (SPFIM) is proposed for PolSAR data. Physical interpretability refers to the interpretability in the physical feature dimension, i.e., the effect of different physical features on the model outputs can be interpreted. In SPFIM, the LSTM is used to process the feature sequences at the pixel level to obtain the physical characteristics importance weights based on the SHAP value. The physical characteristics importance weights are used to weight the original data to obtain new physical-weighted data, which can increase the physical interpretability of the deep learning method.

(2) The SHAP-guided spatial explanation network (SSEN) is proposed, which contains a spatial self-explanation module SSCM based on the Shapley Module [64] design. The spatial SHAP explanation values of the input features can be calculated and can be input as interlayer features to the neural network along with the abstract features. In such a way, the network is allowed to obtain interpretability in spatial dimensions.

The rest of the paper is organized as follows: In Section 2 we present the experimental area, the data used, and the details of the proposed SGEM method. Section 3 provides the experiments and a detailed analysis of the results. A discussion of our work in this study is carried out in Section 4. Finally, Section 5 concludes the paper.

## 2. Materials and Methods

### 2.1. Study Area

The study area is located in Shanwei City, Guangdong Province, China (114°54′W–116°13′24″W, 22°37′40″N–23°38′35″N), as shown in Figure 1. Shanwei City is located in the southeast coast of mainland China, with a total area of 4865.05 square kilometers. It belongs to the southern subtropical monsoon climate zone, with a distinct

maritime climate and abundant light, heat, and water resources. Crops in Shanwei are mainly rice and sweet potatoes, while others include potatoes and corn. The rice cultivation system is a one-year multi-season system dominated by double season rice. The growing season of early rice is mainly from the end of April to August, and that of late rice is from the end of July to December.

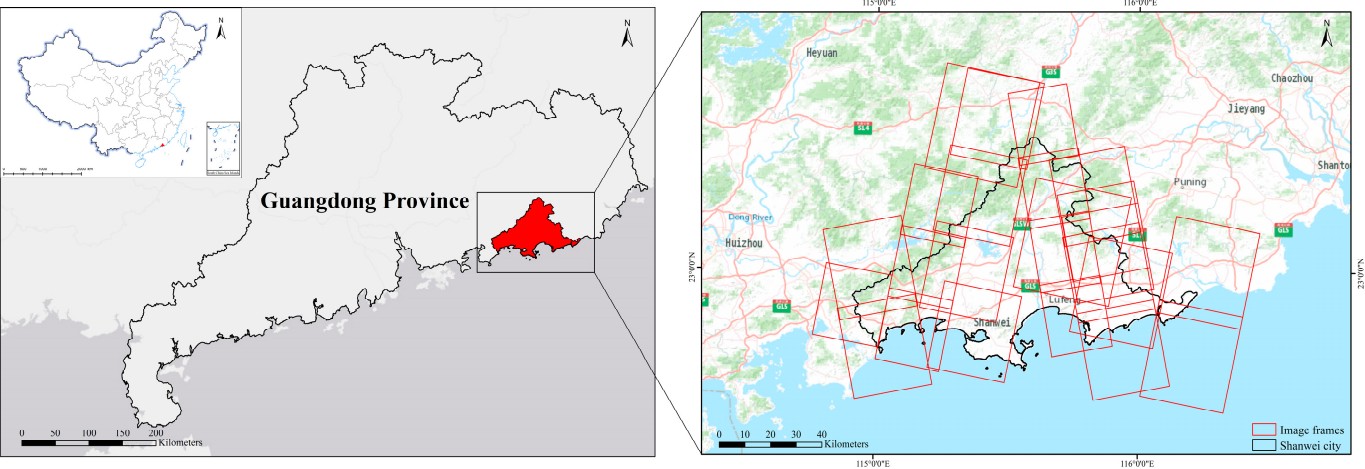

**Figure 1.** Image coverage in Shanwei city, Guangdong Province, China.

### 2.2. Study Data

Launched in 2016, the Gaofen 3 (GF-3) satellite (https://data.cresda.cn, accessed on 25 September 2022) is a C-band, high-resolution PolSAR imaging satellite. In this study, a total of 24 QPSI GF-3 PolSAR images with the resolution of 8 m were obtained, as shown in Figure 1 and Table 1. These time series of PolSAR data covered various phenological phases of rice, which helped to provide physical information of different phenological stages as well as improve the robustness of the model.

**Table 1.** List of GF-3 PolSAR data.

| No. | Date | Incidence Angle | Flight Direction | No. | Date | Incidence Angle | Flight Direction |
|-----|------|-----------------|------------------|-----|------|-----------------|------------------|
| 1 | 30 April 2017 | 35.29°~37.10° | Ascending | 13 | 13 March 2019 | 27.25°~29.84° | Descending |
| 2 | 29 May 2017 | 35.29°~37.10° | Ascending | 14 | 27 May 2019 | 29.70°~31.89° | Descending |
| 3 | 29 May 2017 | 35.29°~37.11° | Ascending | 15 | 27 May 2019 | 29.69°~31.88° | Descending |
| 4 | 24 August 2017 | 35.33°~37.12° | Ascending | 16 | 19 July 2019 | 24.18°~26.81° | Descending |
| 5 | 24 August 2017 | 35.29°~37.11° | Ascending | 17 | 12 February 2020 | 35.62°~37.43° | Descending |
| 6 | 24 August 2017 | 35.29°~37.11° | Ascending | 18 | 12 February 2020 | 35.69°~37.49° | Descending |
| 7 | 10 June 2018 | 29.34°~31.39° | Ascending | 19 | 12 February 2020 | 35.63°~37.44° | Descending |
| 8 | 10 June 2018 | 29.36°~31.42° | Ascending | 20 | 12 June 2020 | 47.07°~48.25° | Descending |
| 9 | 10 August 2018 | 36.84°~38.31° | Descending | 21 | 25 May 2021 | 48.24°~49.26° | Ascending |
| 10 | 10 August 2018 | 36.84°~38.31° | Descending | 22 | 25 May 2021 | 48.21°~49.26° | Ascending |
| 11 | 10 August 2018 | 36.85°~38.32° | Descending | 23 | 25 May 2021 | 48.21°~49.27° | Ascending |
| 12 | 2 January 2019 | 35.29°~37.12° | Descending | 24 | 14 September 2021 | 31.28°~33.41° | Descending |

To collect reliable rice samples for model training, a field survey was conducted in 21 areas of Shanwei City, some examples are shown in Figure 2. With the help of field survey, optical data, ESA WorldCover product [65], and other auxiliary data, rice sample plots were labeled and evenly distributed in Shanwei City.

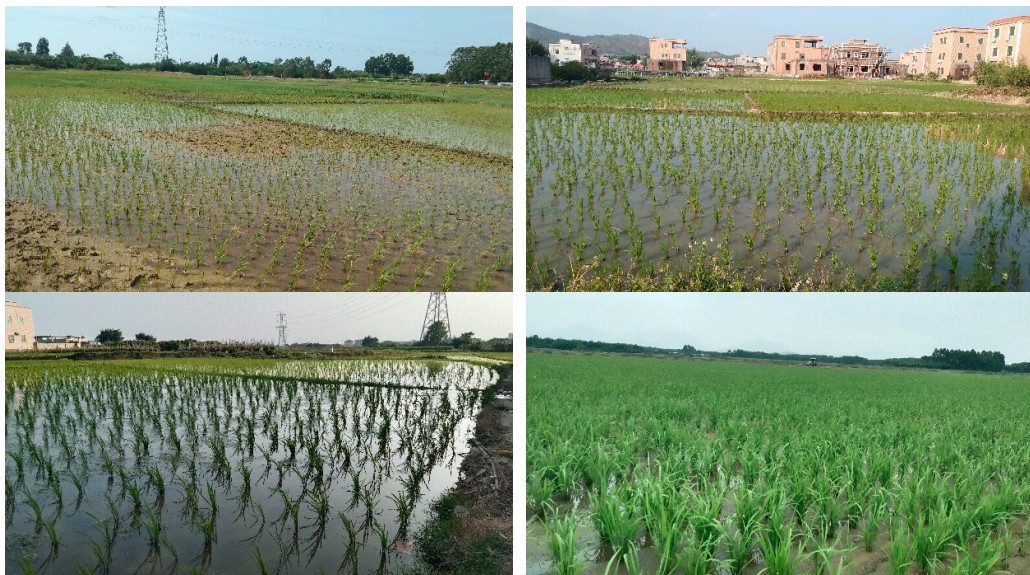

**Figure 2.** Examples of field survey.

*2.3. Methods*

The flowchart of the proposed method is shown in Figure 3. Firstly, the GF-3 PolSAR data were pre-processed, and then Cloude–Pottier decomposition [66], Freeman–Durden three-component decomposition [67], and Yamaguchi four-component decomposition [68] were carried out to extract the physical scattering characteristics. The feature set of rice samples contains 10 physical scattering characteristics.

Then, the proposed SGEM was used to extract the rice distribution. The SHAP feature importance of each feature of the input sample is analyzed by SPFIM to determine the marginal contribution of each input physical characteristic to the final result, and a plausible and intuitive physical explanation of the global SHAP value importance is obtained. The SHAP feature importance is weighted and fused with the original data. Afterwards, rice field extraction is performed by the SSEN. The SSCM is added to the SSEN to calculate the SHAP interpretation values in spatial dimensions, and is used as an inter-layer feature to guide the training and predicting of the network. Finally, the rice distribution map was obtained, and the accuracy of the rice map was verified using field survey.

2.3.1. Physical Features Extraction

At different phenological stages, the polarimetric scattering mechanism of rice is closely related to its growth state. PolSAR data are sensitive to the physical scattering characteristics of targets. Compared with the backscattering coefficients, the polarimetric decomposition methods can specifically describe the scattering mechanism of the target [66–69]. Therefore, the polarimetric decomposition parameters become the basis of the physical interpretability of the model. To characterize the physical information of rice from different perspectives, different decomposition parameters of the target are extracted.

In general, polarimetric decomposition methods can be generally attributed into three categories, i.e., the Huynen-type phenomenological dichotomy [70], the eigenvalue-eigenvector-based decomposition [66,71], and the model-based decomposition [67,68]. Due to the clear physical meaning and simplicity of calculation, the latter two categories have been widely used in scattering mechanism interpretation and land-cover classification over vegetated areas [72,73]. Therefore, in this study the Cloude–Pottier decomposition [66], Freeman–Durden decomposition [67], and Yamaguchi four-component decomposition [68] were selected to calculate the scattering parameters of PolSAR data. Table 2 provides the physical significance of these features.

In this way, a tensor of 10 scattering characteristics of the rice samples could be obtained for model learning and training. These physical characteristics expanded the

feature dimension of SAR data and were used in subsequent physical interpretable analysis and rice field extraction.

### 2.3.2. SHAP-Based Physical Feature Interpretable Module

To evaluate the influence of different polarimetric scattering characteristics in this study and to improve the physical interpretability of deep learning models, the SHAP -based Physical Feature Interpretable Module (SPFIM), shown in Figure 4, was designed to explain the importance of physical characteristics, which is mainly composed of a LSTM model and a DeepSHAP Explainer (DSE). The LSTM mines the connections between different polarimetric scattering characteristics, the DSE interprets the importance of physical characteristics in the form of SHAP values based on the learned LSTM.

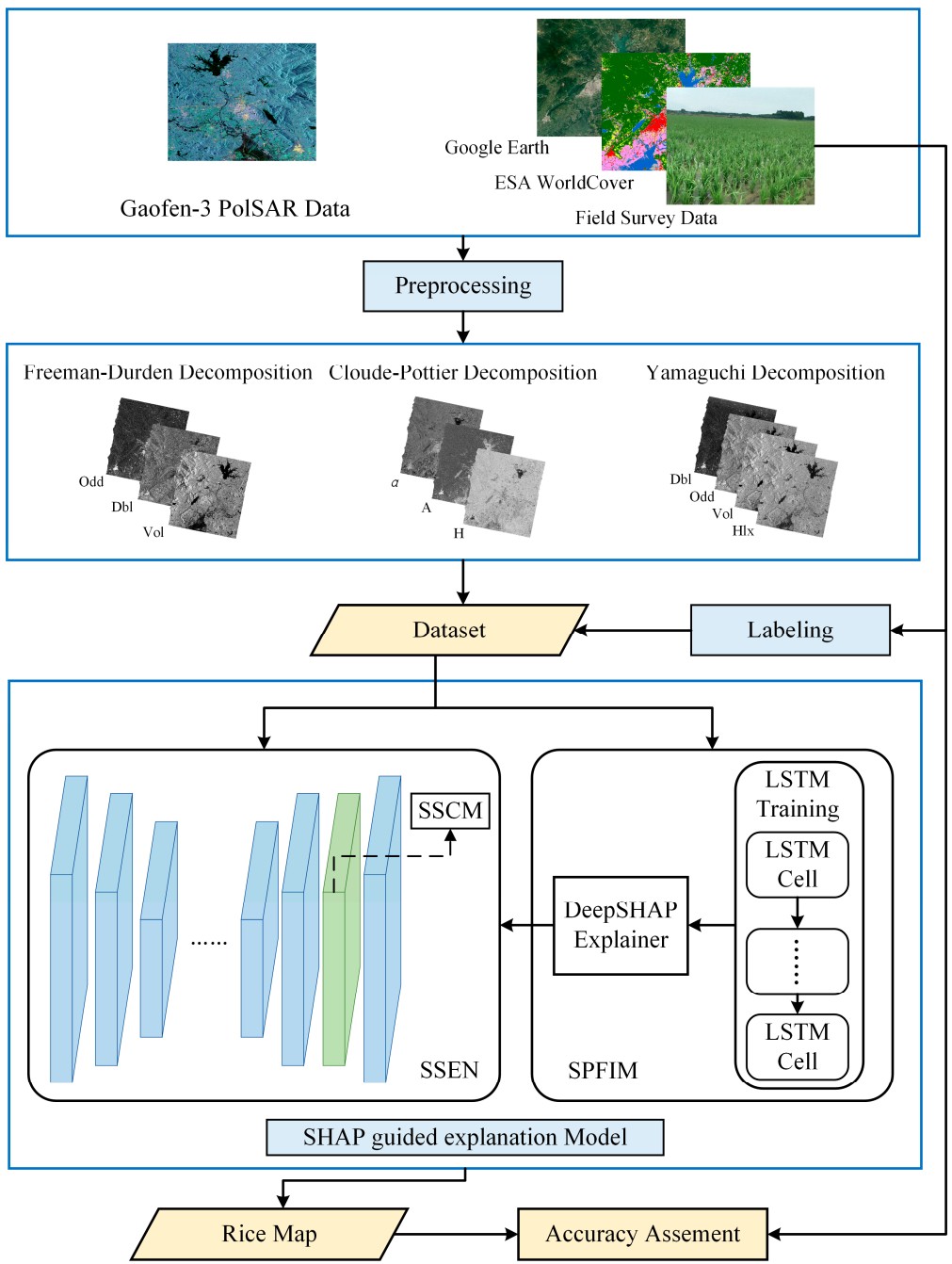

**Figure 3.** The flowchart of the proposed method.

**Table 2.** Characteristic parameters of target decomposition and their physical significance.

| The Characteristic Parameter | Physical Significance |
|---|---|
| $\alpha$ | The size of the average scattering angle $\alpha$ is closely related to the scattering type. $\alpha = 0°$ indicates surface scattering. As $\alpha$ increases, the surface becomes anisotropy. An $\alpha$-value of 45° represents a dipole. If $\alpha$ reaches 90° the scattering process is characterized by dihedral scattering interactions. |
| H | Scattering entropy (H) is an indicator for the number of effective scattering mechanisms, whereby H = 0 belongs to deterministic scattering and H = 1 to totally random scattering. |
| A | Anisotropy (A) only yields additional information for medium values of H. High A signifies that besides the first scattering mechanism only one secondary process contributes to the radar signal. For low A both secondary scattering processes play an important role. |
| Freeman_Odd | Surface scattering of Freeman–Durden decomposition |
| Freeman_Dbl | Dihedral scattering of Freeman–Durden decomposition |
| Freeman_Vol | Volume scattering of Freeman–Durden decomposition |
| Yamaguchi_Odd | Single-bounce of Yamaguchi decomposition |
| Yamaguchi_Dbl | Dihedral scattering of Yamaguchi decomposition |
| Yamaguchi_Vol | Volume scattering of Yamaguchi decomposition |
| Yamaguchi_Hlx | Helix scattering of Yamaguchi decomposition |

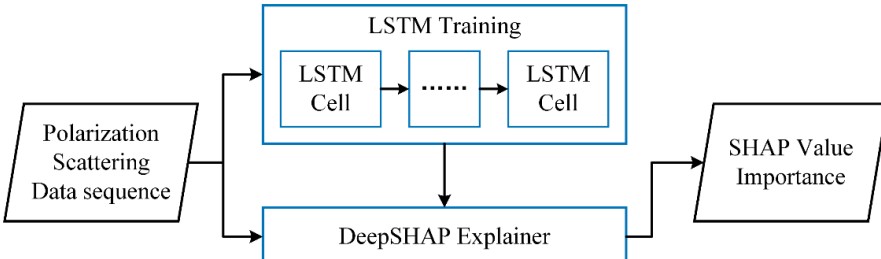

**Figure 4.** Structure of SHAP-based physical feature interpretable module.

LSTM is widely used for sequential data processing tasks such as text analysis and temporal data analysis due to its excellent sequential data processing capabilities. Considering that scattering characteristics are interconnected, i.e., each characteristic has an impact on the model outputs, LSTM is used to fully learn and mine the dependencies between scattering characteristics at the pixel level. Suppose the input rice scattering characteristic slice data are $X \in R^{H \times W \times C}$, with $H$ and $W$ corresponding to the height and width of the slice and $C$ being the characteristic channel dimension of the slice ($C = 10$ in this study). $X$ is first reshaped into a sequence of characteristics in the form of image elements $X' \in R^{(H*W) \times C}$, and then $X'$ is input into LSTM and trained for binary classification, so that it can fully learn and fit the physical characteristics to obtain the sequence model structure and weights.

However, the LSTM itself cannot output an interpretation of the importance of the physical characteristics and DSE is used to interpret the obtained LSTM model.

$f(x)$ represents the trained LSTM model. A single feature sequence $x = [x_1, \ldots, x_C]$ in $X'$ is input to the DSE, and $x$ is first reduced to $x' = [x'_1, \ldots, x'_C]$, where $x'_C \in \{0, 1\}$, and 1 and 0 denote the presence or absence of $x'_C$, respectively. An explanatory model $g(x')$ is then used as an approximation model for $f(x)$ to approximate the SHAP-explained values.

$$f(x) \approx g(x') = \phi_0 + \sum_{i=1}^{D} \phi_i x'_i \tag{1}$$

$\phi_0 = f(h_x(0))$ (all-zero input), and $h_x$ is a mapping function that converts $x'$ to $x$, i.e., $x = h(x')$. Thus, the model output consists of an approximation of the sum of SHAP values for features corresponding to $x_i' = 1$. Then, $g(x')$ is trained to approximate the output of the original network $f(x)$, and the coefficients $\phi_i$ of the model $g(x')$ are used to replace the true SHAP values [48].

DeepSHAP provides an efficient method to estimate the SHAP values of deep neural networks by approximating the absent features with expected values under the assumption of feature independence and model linearity, avoiding the need for repeated model training. Then, the SHAP values of each neural network component (linear, maximum set, and activation) are estimated. The definition in Equation (1) relates SHAP to DeepLIFT [74], which is an additive feature attribution method. The multipliers of DeepLIFT can then be back-propagated to estimate the SHAP values of the model input layer.

After loading the model structure and weights of the LSTM by the DSE, the contribution of the feature is calculated by observing the change in the model output in the presence or absence of a feature, and the final output SHAP interpretation value tensor $Z \in \mathrm{R}^{(H*W) \times C}$. In the tensor $Z$, the SHAP values in the range $[-1, 1]$ of each feature are provided for each input sample x. At this point, the effect of each physical feature on the model results can be clearly understood by statistical analysis of the SHAP values for the purpose of physical interpretability. Further, the physical interpretation tensor is reduced to the size of $H \times W$ with x-weighted fusion, which incorporates the physical interpretation into the data, highlights the important physical information in the data, suppresses irrelevant information, and helps the subsequent rice field extraction.

### 2.3.3. SHAP-Guided Spatial Explanation Network

Learning spatial dimensional relationships is a key focus of CNNs. Semantic segmentation networks often use CNN to extract spatial relationship features, which is a typical "black box" model and lacks interpretability. In order to improve the spatial interpretability of the model, and directly act on the network, the SHAP-guided spatial explanation network (SSEN) is designed to receive the data enhanced by physical explanation for rice field extraction. The structure of SSEN is shown in Figure 5.

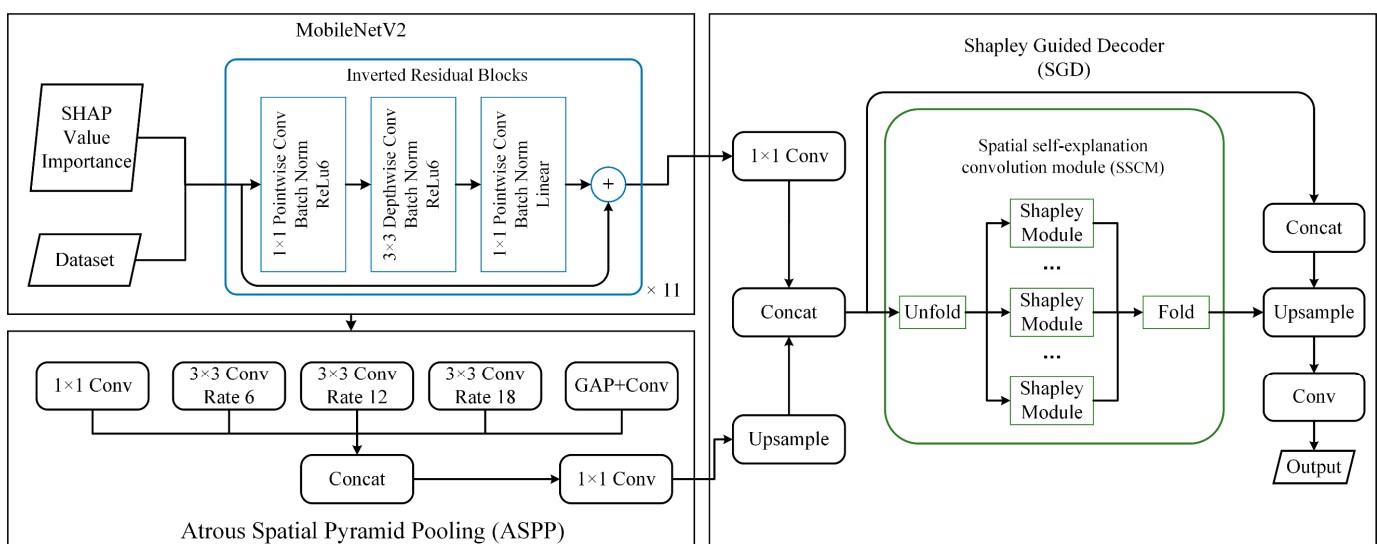

**Figure 5.** Structure of SHAP-guided spatial explanation network.

The network consists of the MobileNetV2 backbone network, the features fusion network of the Atlas Spatial Pyramid Pooling (ASPP), and the spatially self-explaining SHAP-Guided Decoder (SGD). MobileNetV2 is a classical lightweight and efficient feature extraction network, mainly composed of inverse residual linear bottleneck blocks containing depthwise separable convolutions. In the inverse residual linear bottleneck



block, the input tensor is first dimensioned by expansion layer ($1 \times 1$ convolution), then extracted by depthwise layer ($3 \times 3$ convolution), then dimensioned down by projection layer ($1 \times 1$ convolution), and finally the output is activated by linear activation function to avoid the feature loss caused by ReLU6 function. In addition, when the input and output channels are the same, the use of shortcut connection can reuse features and mitigate feature degradation. This structure is used as the feature extraction backbone network to ensure that our model extracts features efficiently with less computational resources. ASPP captures multi-scale information by convolving cavities with different sampling rates in parallel, which avoids the problem of detail information loss of common pooling operations and preserves feature information at different scales to a greater extent. After the network features are extracted and fused, SGD performs upsampling and outputs the results.

In SGD, the SSCM based on the Shapley Module (SM) is designed to make our model spatially self-explanatory, and the basic idea is to use the spatial dimensional SHAP interpretation value as an intermediate layer output feature of the neural network, which directly affects the training and prediction of the model.

SSCM first unfolds the input tensor with a sliding window through the Unfold operation, i.e., the input is cut into patches according to the size of the convolution kernel. Each patch contains the information of neighboring pixels within the convolution kernel. Then, the patches are fed into a parallel SM instead of the matrix multiplication operation of traditional convolution. To relieve the computational pressure, each SM takes only some of the input features at a time to compute the SHAP values, and the final output depends on all the computed SHAP values, and the operation is represented as follows.

$$\Omega(x, f) = [\phi_1(f, x_1), \ldots, \phi_d(f, x_d)]^T \tag{2}$$

$$g(x) = \text{sum}(\Omega(x, f)) \tag{3}$$

where $\phi_d(f, x)$ denotes the SHAP value computed by the model function f on a subset $x_d$ of the inputs, sum denotes the summation, and $g(x)$ is the final SHAP-explained value.

After obtaining the output tensor corresponding to the original pixel, multiple values at each pixel are summed and finally reduced to a SHAP tensor with the same dimensional size of the input tensor space by the Fold operation, which assigns a SHAP interpretation value to each input pixel.

The original SHAPNET replaced the interlayer output with SHAP interpreted values. In contrast, the purpose of SSCM is not to completely replace the interlayer representation of the network with SHAP interpreted values, but to serve as an interpreted reference to assist in network decision making. For this reason, SSCM is placed at the end of the network structure to evaluate the importance of the spatial relationships of the feature map pixels after the feature extraction and fusion between the backbone network and ASPP is completed, and splicing with the input data into the next layer of the network for spatial self-explanation.

### 2.3.4. Accuracy Assessment

This study measures the performance of the method based on accuracy metrics such as overall accuracy, precision, recall, *F*1, and kappa calculated from the confusion matrix. Their definitions are as follows:

$$\text{Overall Accuracy} = \frac{TP + TN}{TP + TN + FN + FP} \tag{4}$$

$$\text{Precision} = \frac{TP}{TP + FP} \tag{5}$$

$$\text{Recall} = \frac{TP}{TP + FN} \tag{6}$$

$$F1 = \frac{2TP}{2TP + FP + FN} \tag{7}$$

$$\text{Kappa} = \frac{\text{accuracy} - P_e}{1 - P_e} \tag{8}$$

$$P_e = \frac{(TP + FP) \times (TP + FN) + (FN + TN) \times (FP + TN)}{(TP + TN + FN + FP)^2} \tag{9}$$

where $TP$ is the number of correctly recognized rice pixels; $TN$ is the number of correctly recognized non-rice pixels; $FP$ is the number of non-rice pixels incorrectly represented as rice; $FN$ is the number of rice pixels ignored and marked as non-rice; and $P_e$ is the expected precision.

## 3. Experiments and Results

The method proposed in this study was mainly based on TensorFlow, PyTorch, and SHAP. All calculations were executed on a Windows 10 workstation using NVIDIA GeForce RTX 3090 GPU. The training batch size of the physical feature importance analysis module was 64, the learning rate was initialized to 0.001 and adjusted according to the training time, using Adam optimizer and cross-entropy loss function. The training batch size of the rice field extraction network part was 32, the learning rate was initialized to 0.0025 and adjusted with training time, and the SGD optimizer and cross-entropy loss function were used.

In order to verify the interpretability and reliability of the proposed model, three experiments were conducted: (1) verification of the physical interpretability of the model; (2) verification of the spatial interpretability of the model; (3) comparison of the performance of the proposed method with other methods.

### 3.1. Physical Interpretability of SSEN

To verify the physical interpretability of the proposed method, five periods were selected for physical characteristics importance analysis in the images from April to August, which basically covered the complete growth period of rice. Three main stages of rice are usually defined for the rice growth cycle, namely, vegetative, reproductive, and maturation or ripening.

Firstly, the LSTM was trained to fit the training set. Then, using the trained LSTM model, the sliced data involving the five periods in the test set were input into SPFIM to obtain the global SHAP feature importance bar graphs and local SHAP feature importance scatter plots about the five periods, each set of images consisted of 1000 randomly selected samples. As shown in Figure 4, the SHAP feature importance bar chart showed the average impact of each sample from a global perspective, and the SHAP feature importance scatter plot represented the SHAP value of each sample and the magnitude of its own value from a local perspective.

On 30 April the rice was in the seedling stage or transplanting stage, belonging to the vegetative phase. At this stage, the rice seedlings were short and sparsely distributed, and the underlying surface was mostly water or wet soil. The microwave first reflected with the underlying surface, and then dihedral scattering occurred with the rice stem, so the SHAP value of dihedral scattering calculated by Yamaguchi decomposition and Freeman–Durden decomposition was high. From the scatter plot, it can be found that the average scattering angle $\alpha$ below 45° had a low contribution to the model output, i.e., the diffuse reflection of the rough surface had a low contribution to the model output, and the scattering angle value representing the volume scattering had a positive contribution to the model. Overall, the dihedral scattering component decomposed by Freeman–Durden and Yamaguchi were of the highest importance, indicating that dihedral scattering was dominant at this time, which was consistent with the work of Lopez-Sanchez et al. [15].

On 29 May the rice was in reproductive stage. Rice plants became taller, and the volume scattering and scattering randomness of the canopy increased, indicating the coexistence of multiple scattering mechanisms. However, the volume scattering was less than the dihedral scattering and surface scattering. As can be seen from the Figure 6, the SHAP importance of Freeman's volume scattering component increased to the second

position, and most of the samples provided positive contributions. Dihedral scattering component in Yamaguchi's decomposition had a large influence on the model output, because the rice canopy was not fully developed. After a 14-day interval, the rice plant and canopy were further developed by June 12, and the higher values of volume scattering and dihedral scattering provided more positive contributions.

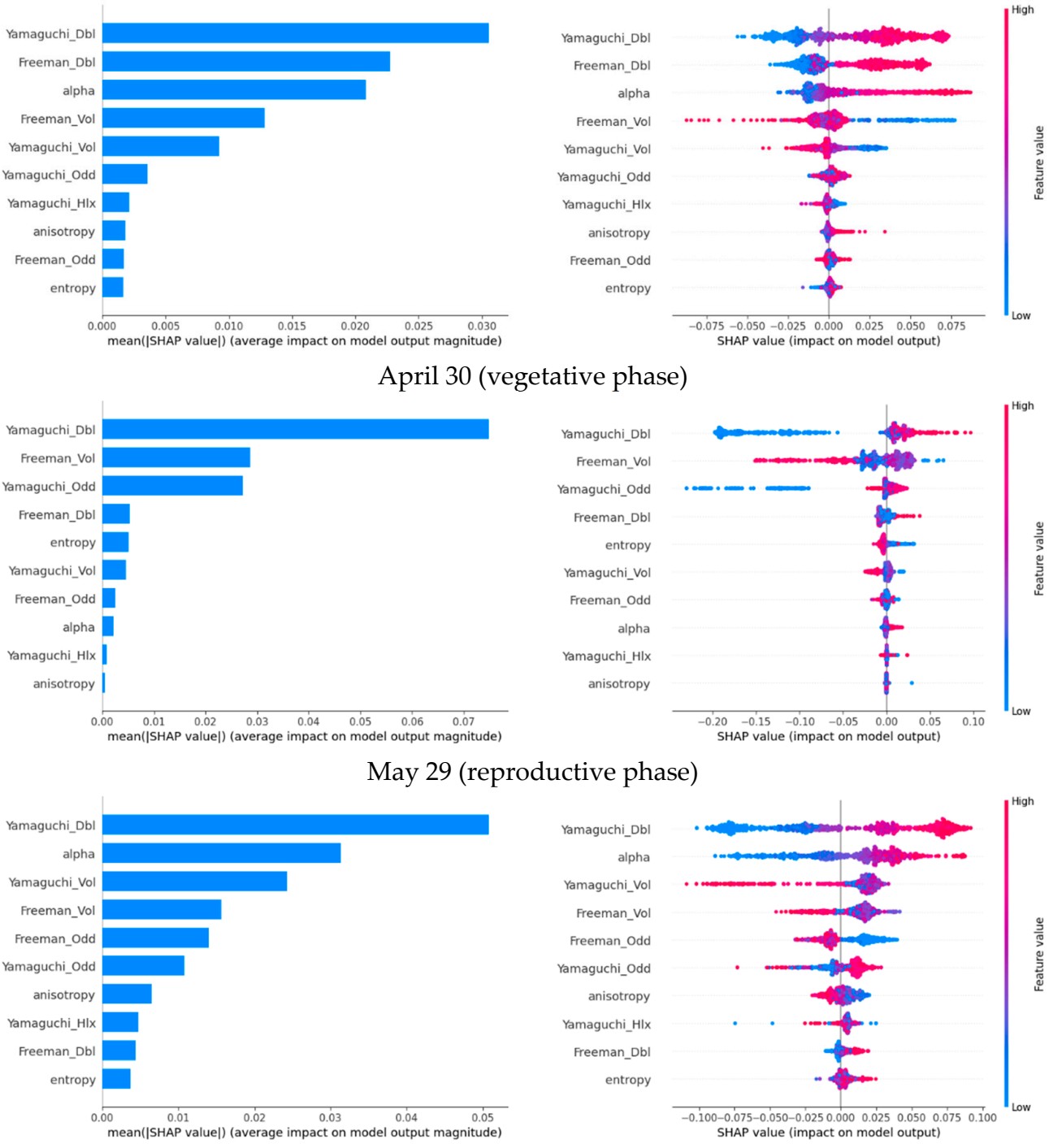

April 30 (vegetative phase)

May 29 (reproductive phase)

June 12 (reproductive phase)

**Figure 6.** *Cont.*

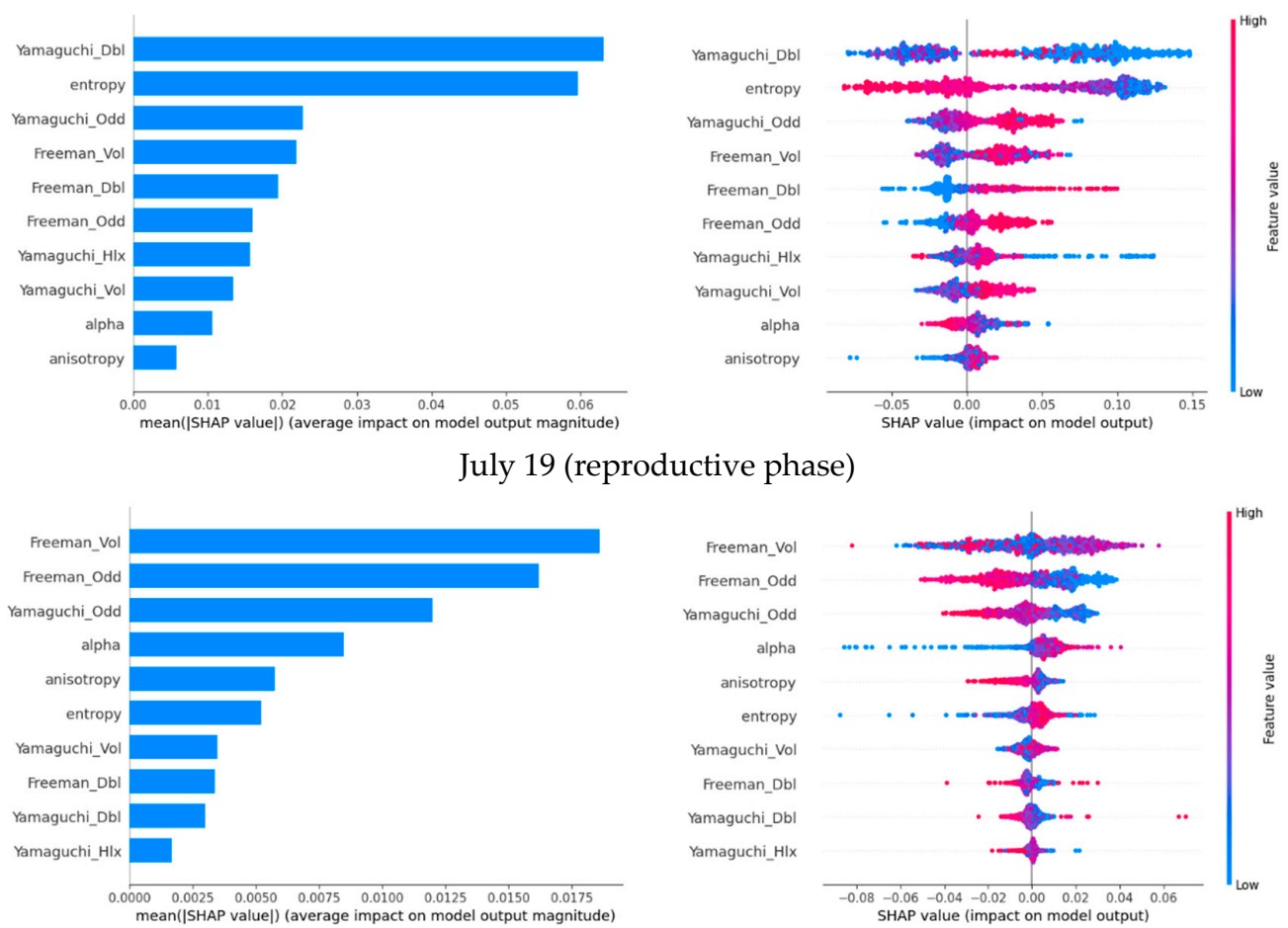

July 19 (reproductive phase)

August 10 (maturation phase)

**Figure 6.** Physical feature importance histograms and scatter plots.

On 19 July the rice grew to the pollination stage of reproductive phase, when it responded to more diverse physical characteristics, and the importance of each physical characteristic increased. Dihedral scattering maintained a high importance, but the distribution of eigenvalues was more dispersed and the overall values began to decrease; the influence of scattering entropy increased because the polarization state of rice was more random at this time, but too much randomness was not conducive to rice field extraction. It should be noted that the importance of surface scattering for Freeman decomposition and Yamaguchi decomposition started to increase, while the volume scattering started to decrease, which might be caused by the inconsistent rice phenology in different areas.

On 10 August when the rice was at maturation phase, it was ripe with higher volume scattering and significantly weaker dihedral scattering. In addition, the importance of surface scattering was obviously higher, possibly because some rice fields had been harvested.

Overall, the SHAP importance values calculated by the physical feature interpretation module for samples of different periods can better reflect the association between rice physical characteristics and phenological periods and can help users intuitively understand the role of physical features in the operation of the model and understand the driving factors for decision generation. It is worth emphasizing that the SHAP-explained values respond to the importance of the features in the model, and their trends are different from those of the actual characteristic values.

### 3.2. Spatial Interpretability of SSEN

Paddy rice fields show a large concentrated and continuous distribution in space. Since spatial dimensionality is an essential learning direction of convolutional networks,

deep learning models based on convolutional neural networks are often utilized to retrieve the spatial distribution characteristics of rice. The proposed SSCM can internalize the interpretability of the spatial dimension into the neural network, change from post-hoc interpretation to inter-layer self-interpretation of the neural network, improve the interpretability of the deep learning model in the spatial dimension, and apply the potential interpretation directly to the network itself.

To reflect the spatial self-interpretation of the proposed method, two test regions were selected to compare the interlayer characteristics of SGEN without and with SSCM for the same number of training rounds (200 rounds), as shown in Figure 7 which visualized the effect of using spatial interpretation as the potential interpretation of the interlayer.

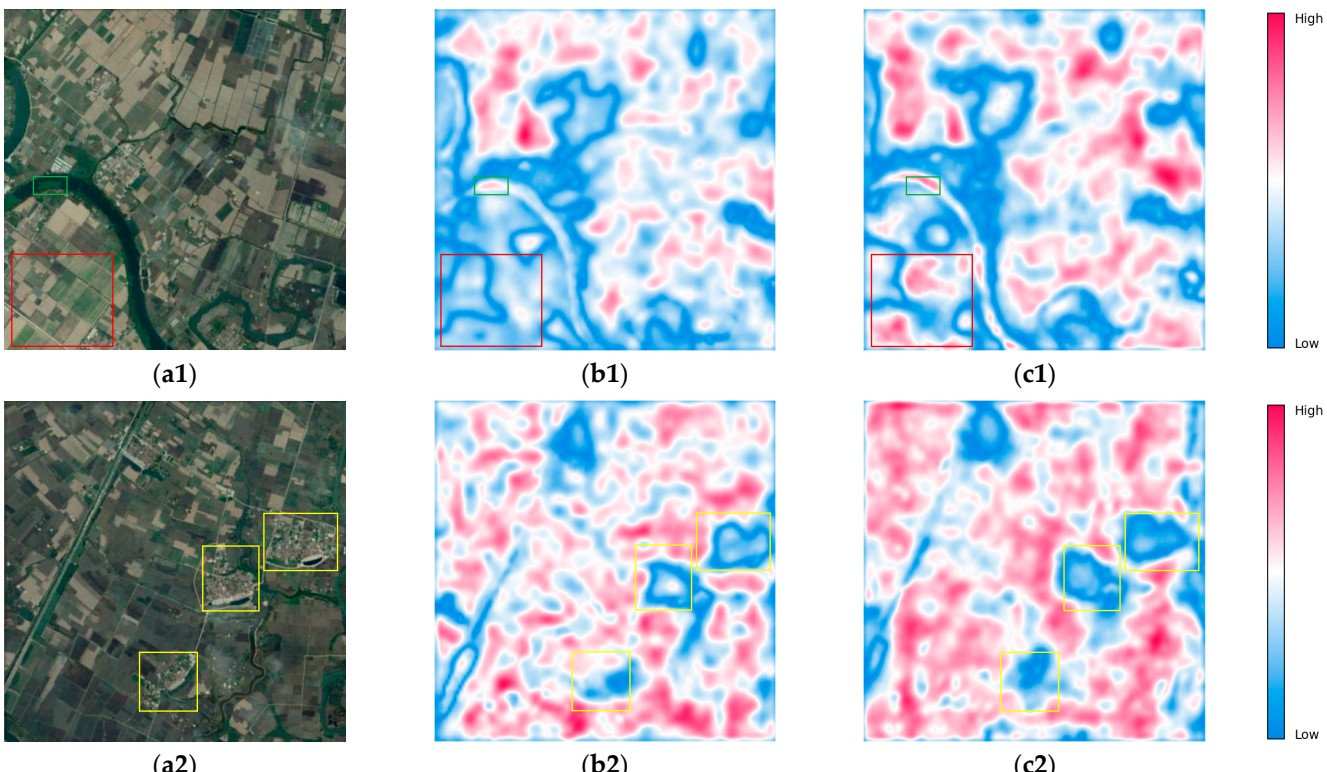

**Figure 7.** Comparison of the interlayer characteristics of SGEN without and with SSCM: (**a1,a2**) are the optical images (13 March 2018) of different rice regions, respectively; (**b1,b2**) are the interlayer characteristics of SGEN without SSCM; (**c1,c2**) are the interlayer characteristics of SGEN with SSCM. The red, green, and yellow borders indicate the rice area, non-rice area, and building area, respectively.

As can be seen in Figure 7, for the same number of training sessions, the model with SSCM focused on a larger and more uniformly distributed rice areas (e.g., Figure 7(c1,c2)). In contrast, when SSCM was not added, the rice areas that the model mainly focused on were characterized by small areas and were discontinuously distributed, which obviously did not fit the actual rice distribution characteristics (e.g., Figure 7(b1,b2)). As can be seen from the optical image shown in Figure 7(a1), some rice fields were also present on the left side of the river (indicated by the red border), but the model did not show significant attention to this region when the SSCM was not included (Figure 7(b1)). Moreover, Figure 7(c2) shows that the model paid significantly less attention to building areas, indicating that the model could effectively distinguish between building and rice. These results show that the SSCM allows the model to accurately locate rice areas and achieves accurate differentiation between rice and non-rice.

### 3.3. Comparison of Different Methods

As we know, Random Forest (RF) is the more classical machine learning classification algorithm that is widely used for SAR rice field extraction [75–77], and Deeplabv3+ is the most advanced deep learning semantic segmentation algorithm available. Therefore, in this section, the performance of the proposed algorithm was verified by comparing SGEM with RF and Deeplabv3+ model.

Based on the field survey, the accuracy of the three models is shown in Table 3. The proposed SGEM method obtained the highest overall accuracy (up to 95.73%). Its overall accuracy was 5.97% and 2.72% higher than RF and Deeplabv3+, respectively. The Precision, F1, and kappa of SGEM model were also higher than RF and Deeplabv3+. Although Deeplabv3+ model had a higher recall than SGEM model, its accuracy was significantly lower than that of SGEM, which meant that more non-rice was incorrectly identified as rice. These results indicate that the SGEM model significantly outperforms the RF and Deeplabv3+ models.

**Table 3.** Accuracy evaluation of three models.

| Model | Overall Accuracy | Precision | Recall | F1 | Kappa |
|:---:|:---:|:---:|:---:|:---:|:---:|
| RF | 89.76 | 89.69 | 91.41 | 90.54 | 0.7939 |
| Deeplabv3+ | 93.01 | 90.31 | 97.45 | 93.74 | 0.8586 |
| SGEM | 95.73 | 97.15 | 94.82 | 95.97 | 0.9143 |

In addition, to reflect the advantages of SGEM, areas with concentrated rice distribution were selected for detailed comparative analysis, as shown in Figure 8. As RF only learns the classification at pixel level, its ability to learn the physical sequence information is limited. As can be seen from Figure 8b, there were many missing areas in the extraction result, and the extracted land parcels were too fragmented. The extraction results of Deeplabv3+ are shown in Figure 8c. This method produced fewer missing areas and more concentrated rice distribution, but the details of the overall rice field boundaries and interior were not finely carved and there were more false detections. Compared with the classification results of Deeplabv3+ and RF, the classification results of SGEM in Figure 8d had the fewest missing areas, and the overall plot boundaries were more regular, and the details of rivers and roads between fields were finely carved (as shown in the red border in Figure 8d). The results show that the proposed method has better recognition ability for rice and non-rice.

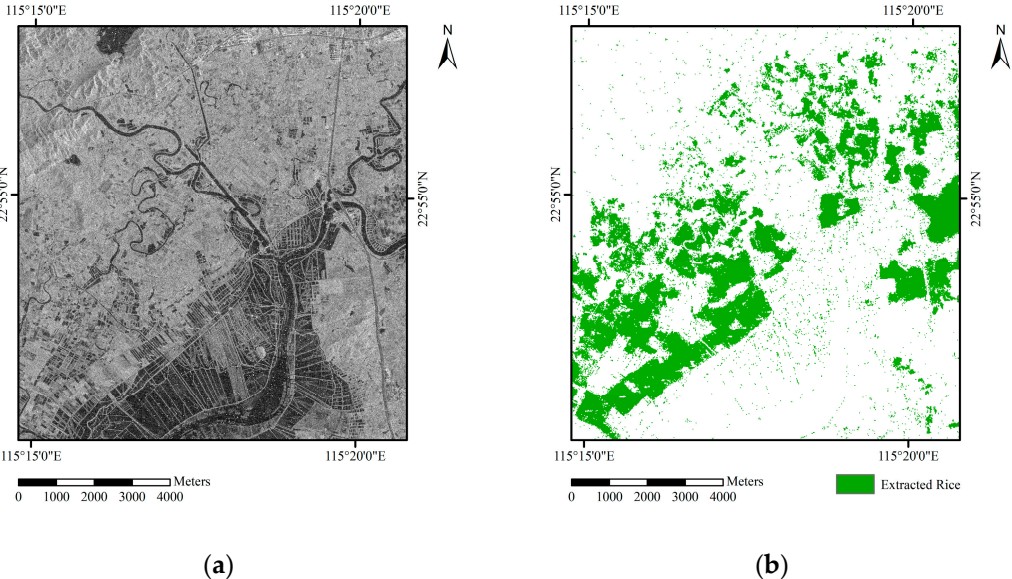

(a)　　　　　　　　　　　　　　　　　　　　　　　　　　　　(b)

**Figure 8.** *Cont.*

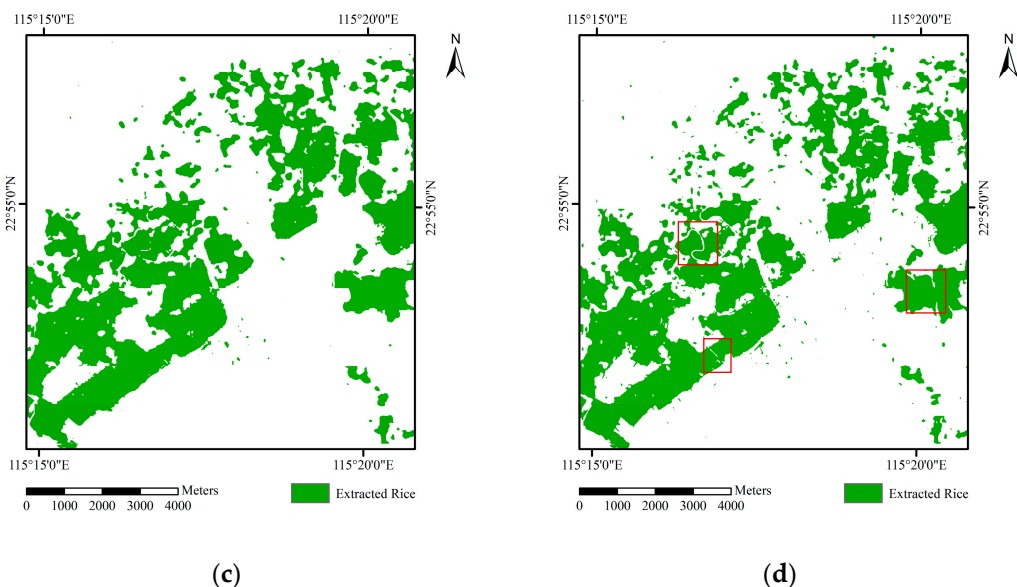

**Figure 8.** Comparison of rice field extraction results by different methods: (**a**) SAR; (**b**) RF; (**c**) Deeplabv3+; (**d**) SGEM.

## 4. Discussion

The SGEM method proposed in this study achieved the internalization of physical and spatial interpretations into the network itself. Using PolSAR data, the Cloude–Pottier decomposition, Freeman decomposition, and Yamaguchi decomposition were used to characterize the physical characteristics of rice, and the SPFIM method was able to provide the physical feature interpretation associated with the crop phenology, improving the physical interpretability of the method. The importance of physical interpretation was obtained based on the DeepSHAP method and combined with original data, which was a process of internalizing the physical interpretation. In this way, a new dataset containing physical interpretation was obtained, which effectively suppressed the noise in the dataset and optimized the data distribution. In the rice field extraction stage, the SGCM method was used to innovatively internalize the spatial SHAP interpretation values into the network as inter-layer output, transforming the post-hoc interpretation into self-interpretation, enabling the network to rely on spatial self-interpretation to optimize the parameters during the training process. As a result, the proposed model can converge faster and obtain better segmentation details. Compared with other methods, the SGEM method obtains better rice region extraction results and, more importantly, has good physical and spatial interpretation capabilities, the interpretability of the whole method is greatly improved, and the interpretation results can be directly applied to the model training and prediction process.

The method also has limitations. In Figure 7(a1), some vegetation (green box area) seems to be in the river, so Figure 7(b1,c1) both focus some attention to the river interior, which may lead to false alarms. In the future, the method performance is further improved by increasing the negative samples in the water area. In addition, to reduce computational effort and improve computational efficiency, overall physical feature SHAP values are assigned to each slice input during physical interpretation. In fact, the SHAP values of the physical characteristics corresponding to each pixel are available, thus allowing a refined portrayal of the importance of the physical features of each pixel.

## 5. Conclusions

To solve the problem of insufficient physical interpretability and spatial interpretability of the deep learning SAR rice field extraction algorithm, a new SHAP value-guided interpretable model is proposed in this study, and high-precision rice field extraction is achieved using GF-3 PolSAR data. To address the problem of insufficient physical interpre-

tation, SPFIM was designed to analyze the importance of physical features at the pixel level with the help of LSTM sequence model and SHAP values to provide a plausible physical interpretation, and combined with the original data weighting to make the physical interpretation internalized into the data; after that, a SHAP-guided spatial explanation network, consisting of MobileNetV2 backbone network, ASPP, and SHAP-Guided Decoder (SGD) was constructed, and the SGD contains a spatial self-explanation convolution module that translates the spatial interpretation of SHAP values into interlayer features of the network so that the network obtains the spatial self-explanation property and uses it to optimize itself. The experimental results in Shanwei City, Guangdong Province, China as the study area show that the physical interpretation provided by this method has high correlation with the rice phenology, and the spatial self-explanation feature can make the extracted rice field details richer, and the overall accuracy of the rice mapping results is 95.73%, and the kappa coefficient reaches 0.9143. Compared with other methods, it has high interpretability and practical value.

**Author Contributions:** Conceptualization, methodology, software, J.G., C.S. and H.Z.; validation, formal analysis, J.G. and H.Z.; investigation, C.S. and L.X.; resources, data curation, C.S. and J.G.; writing—original draft preparation, J.G. and H.Z.; writing—review and editing, H.Z., L.X. and J.G. and Z.G.; visualization, L.X. and H.D.; supervision, project administration, H.Z. and C.W. All authors have read and agreed to the published version of the manuscript.

**Funding:** This research was funded by the National Natural Science Foundation of China under Grants 42001278, 41971395, and 41930110.

**Data Availability Statement:** The authors do not have permission to share data.

**Acknowledgments:** We sincerely thank the anonymous reviewers for their critical comments and suggestions for improving the manuscript.

**Conflicts of Interest:** The authors declare no conflict of interest.

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
