# Peer review of "A Physically Interpretable Rice Field Extraction Model for PolSAR Imagery"

_remotesensing, doi:10.3390/rs15040974_

Round 1

Reviewer 1 Report

Comments can be found in the attachment.

Reviewer 2 Report

This paper proposes a novel SHAP value-guided interpretable model for the rice field extracion task. Generally the paper is well written and has relatively high novolty. Before being accepted, the following point should be cleared:

1. The paper compares SGEM weith Deeplabv3+ and RF, which is relatively weak to convince readers. Is there any other method that focus on this task? Can the author compare or qualitatively analysis the performance of SGEM with other scheme methods? For example, the Unet-based methods?

2. The paper says : the  scattering characteristics are interconnected, can the author explain more about this point? 

Reviewer 3 Report

Dear Authors,

I have included my comments to the PDF file of the manuscript. Authors are required to follow the comments to revise their manuscript.

Good luck.

Reviewer 4 Report

The paper "A Physically Interpretable Rice Field Extraction Model for Pol-2 SAR Imagery" focuses on the development of a new SHApley Additive exPlanation (SHAP) value-driven explanation model for polarimetric SAR data. The physical characteristics of rice were extracted using polarimetric scattering decomposition, and a SHAP-guided spatial interpretation network (SSEN) was constructed to internalize the spatial interpretation values in the network layer to optimize the spatial refinement of the extraction results. The authors propose two new physical interpretability and classification methods: SPFIM and SSEN. They used GAOGEN 3 data obtained between 2017 and 2021 and rice samples. The physical characteristics of SGEM were compared with Random forest and DEEPlabv3+. The overall accuracy of rice mapping results was 95.73% for SGEM with a kappa coefficient of 0.9143.

Recommendations for authors: typo in line 229 .

When was the satellite data from figure 7 acquired?

Since the article refers to polarimetric data, I would like to see how SGEM is applied to polarimetric data, possibly using the same window as in figure 7.

The results from this study are not compared with other studies in which SHAP was used. Even though the algorithm is new, based interpretable object detection method has been used in many other studies.
